# Evaluation of upper-tropospheric lower-stratospheric properties over the Asian monsoon region in a storm-resolving model

Sylvia C. Sullivan[1,2], Aiko Voigt[2], Edgardo Sepúlveda Araya[1], Silvia Bucci[2], Annette Miltenberger[3], Meredith K. Kupinski[1,4], Christian Rolf[5], and Martina Krämer[3,5]

[1]Department of Chemical and Environmental Engineering, University of Arizona, Tucson, Arizona, USA
[2]Department of Meteorology and Geophysics, University of Vienna, Vienna, Austria
[3]Institute for Atmospheric Physics, Johannes Gutenberg University, Mainz, Germany
[4]Wyant School of Optical Sciences, University of Arizona, Tucson, Arizona, USA
[5]Forschungzentrum Jülich, Jülich, Germany

**Correspondence:** Sylvia C. Sullivan (sylvia@arizona.edu)

**Abstract.** The structure of the tropical upper troposphere-lower stratosphere (UTLS) affects radiative balance, stability, and regional dynamics in important ways. Lack of reliable observational baselines poses a challenge to evaluating model representation of UTLS properties. Here, we use in-situ data, primarily from the StratoClim field campaign over the Asian Monsoon area, to assess the UTLS temperature, moisture, and ice clouds in the Icosahedral Nonhydrostatic (ICON) model at storm-resolving grid spacing. We also employ superpressure balloon data and updrafts of the POSIDON and ATTREX campaigns to evaluate the UTLS convective updrafts and gravity wave activity in ICON. Our simulations show the upper troposphere is too cold, while the lower stratosphere is too warm and excessively dry relative to observations. These thermodynamic biases coincide with overestimated cloud ice in the upper troposphere and underestimated cloud ice in the lower stratosphere. The mean convective updraft is underestimated by 80% in the model, and the power spectral density for temperature fluctuations of frequencies greater than $10^{-3}$ s$^{-1}$ is underestimated by orders of magnitude. Too weak dynamics exacerbate a lack of ice cloud above 100 hPa. Too weak and too infrequent convective overshoots or too rapid dissipation of upper-tropospheric ice clouds in the model are two possible explanatory mechanisms for these biases.

## 1 Introduction

In the tropics, atmospheric properties transition from those of the troposphere to those of the stratosphere over a few kilometers in the *upper-troposphere lower-stratosphere* (UTLS). Accurate representation of UTLS thermodynamics and cloud structure is critical for several reasons. First, warmer minimum temperatures in the UTLS allow more moisture to enter the stratosphere, since higher temperatures raise the minimum saturation vapor pressure of ascending air. Stratospheric moisture can, in turn, intensify the warming from carbon dioxide emissions by 0.1-0.3 W m$^{-2}$ K$^{-1}$ because of the strong infrared absorption of water vapor (Banerjee et al., 2019; Dessler et al., 2013). This feedback can alter globally-averaged surface temperatures non-linearly, for example with a 10% decrease in stratospheric water vapor leading to a 25% decrease in the surface warming rate (Solomon et al., 2010). Second, UTLS stability determines characteristics of convective overshoots and anvil cloud outflow

from convection. Sharper gradients in potential temperature correspond to greater stability and suppress overshoot occurrence and strength, as well as anvil extent. Finally, the radiative properties of UTLS moisture and clouds affect dynamics. For example, increased stratospheric moisture can cause poleward shifts in the storm tracks due to differential radiative effects between the midlatitudes and tropics (Maycock et al., 2013; Charlesworth et al., 2023). Incorrect representation of UTLS ice clouds can also disrupt the description of feedbacks between these clouds and gravity wave-generated turbulence (Podglajen et al., 2017).

Existing evaluation studies find pervasive but inconsistent UTLS biases across models. For example, in regard to temperature biases, a comparison of 13 different chemistry-climate models found a 10-K spread in tropical tropopause temperatures (Gettelman et al., 2009). The CMIP5 models exhibited too-warm anomalies near the tropopause (Kim et al., 2013), while too-cold anomalies have been documented near the tropopause for the Icosahedral Nonhydrostatic (ICON) model used here (Crueger et al., 2018). More recently, Merlis et al. (2024) evaluated changes in the vertical distribution of temperature using both CMIP6 global climate models (GCMs) and the X-SHiELD GSRM under +4-K sea surface temperature or 4xCO2 perturbations. Already in the control climatology, the average tropopause temperature bias in X-SHiELD was ∼6 K, 4 K higher than that in the CMIP6 GCMs. In regard to moisture structure, Gettelman et al. (2010) reported too much moisture above the cold point across 18 chemistry-climate models. More recently, Ploeger et al. (2024) documented too much water vapor over the Pacific UTLS across many CMIP6 models. UTLS ice cloud biases also exist; for example, nudged GSRMs robustly overestimate upper-tropospheric cloud fraction and underestimate its altitude (Atlas et al., 2024).

Importantly, anomalies in UTLS moisture are often largest for the Asian monsoon area (AMA) (e.g., Schoeberl et al., 2013; Ploeger et al., 2024), and multiple factors complicate the realistic model representation of UTLS thermodynamic and cloud structure in this region. Cloud overlap complicates the vertical profiles of UTLS radiative fluxes and heating (Johansson et al., 2019). The fine-scale features of UTLS moisture and temperature are not well-captured by the limited vertical grid spacing of most models (e.g., Marécal et al., 2007; Wang et al., 2019), and validating measurements are often limited for this region (e.g., Gettelman et al., 2010). In regard to this last challenge, flying above 15 km poses a challenge for many aircraft, given the very low temperatures and pressures there. The water vapor mixing ratios of the tropical UTLS are very low, requiring high instrument accuracy (Singer et al., 2022). Reanalysis data do not reliably capture the UTLS structure, with different reanalyses predicting variations in tropopause height of 100–300 m and in tropopause temperature of ∼1.5 K, despite improvements over time (Hoffmann and Spang, 2021). Known biases existed in the representation of the UTLS region in the ERA-5 reanalysis, including underestimates of lower stratospheric temperature and overestimates of lower stratospheric moisture due to overactive mixing (e.g., Krüger et al., 2022; Simmons et al., 2020). Several challenges exist also for satellite measurements: Microwave sounders lack vertical resolution, while moisture retrieval from radio occultation requires additional constraints and infrared sounders are susceptible to cloud contamination.

In this study, we make use of a unique in-situ dataset of UTLS properties to evaluate their representation in the ICON model at storm-resolving grid spacing ($\Delta x = 2.5$ km). Dauhut and Hohenegger (2022) have evaluated the UTLS moisture *budget* in a global storm-resolving model (SRM) for the first time, but most recent studies evaluating UTLS structure or chemistry still use coarse grid spacing (e.g., Smith et al., 2024; Cohen et al., 2025). Exceptions are studies of Nugent et al. (2022) and Turbeville et al. (2022) which assess tropical ice cloud properties across several SRMs that participated in the DYAMOND

model intercomparison against satellite data. We build upon these studies but consider multiple aspects of UTLS structure—thermodynamic and dynamic fields, as well as cloud properties—in the ICON model. We also use in-situ measurements of the Stratospheric and upper tropospheric processes for better climate prediction (StratoClim) and Strateole-2 field campaigns, rather than satellite data, for our evaluation. StratoClim took place out of Kathmandu, Nepal in August 2017 and measured temperature, specific humidity, trace gas concentrations, and cloud and aerosol particle size distributions in the AMA UTLS, while the first campaign of Strateole-2 released several long-duration superpressure balloons from the Seychelles in February 2020. We finally use vertical velocity measurements of the POSIDON campaign in October 2016 and the ATTREX campaign in early 2014, both over the west Pacific and detailed in Section 2.1. With these data, we investigate how well (or not) ICON can represent temperature, moisture, vertical velocity, and ice cloud structures in the AMA UTLS.

## 2 Methods

### 2.1 Field Campaign Measurements

We compare our synthetic flight tracks to measurements from the M55 Geophysica aircraft during StratoClim Flight 7, which observed interesting UTLS moisture and ice cloud structure. StratoClim had the broad goal of better understanding dynamics, microphysics, and chemistry in the upper troposphere lower stratosphere (UTLS) of the Asian monsoon region (www.stratoclim.org). Flight 7 took place from 04:30 to 06:50 UTC on 8 August 2017 between Kathmandu and west Bengal (red trace in Figure 1a). Two instruments aboard the Geophysica measured water vapor concentrations: the Fluorescent Lyman-$\alpha$ Stratospheric Hygrometer for Aircraft (FLASH-A) and the Fast In-situ Stratospheric Hygrometer (FISH) (Khaykin et al., 2013; Meyer et al., 2015). Ice water content is then calculated as the difference of total water (both gas-phase and evaporated ice crystals) measured by FISH and gas-phase water measured by FLASH-A (Afchine et al., 2018). Measurement frequency was 1 Hz with an uncertainty of 6% for FLASH-A measurements and 6-8% for FISH measurements (Lee et al., 2019). To prevent measurement contamination by very moist tropospheric air, FLASH-A only measures at pressures below 250 hPa and FISH only measures below 350-400 hPa. When FISH and FLASH-A measurements were not simultaneously available, ice water content was inferred from particle size distribution measurements of the NIXE-CAPS cloud spectrometer (Krämer et al., 2020).

Vertical velocities were not measured during StratoClim, so we compare our simulated ascent rates to those from the Pacific Oxidants, Sulfur, Ice, Dehydration, and cONvection (POSIDON) campaign in October 2016 and the Airborne Tropical TRopopause EXperiment (ATTREX) in January and February 2014, both based out of Guam (Jensen et al., 2017). POSIDON and ATTREX both studied the western Pacific tropopause moisture and chemical composition and measured vertical velocities with a Meteorological Measurement System (MMS), with POSIDON sampling more often near deep convection. Global statistics of vertical velocity are only beginning to emerge with novel retrievals and remote sensing techniques (e.g., Dolan et al., 2023; Poujol and Bony, 2024), so we rely on in-situ data as geographically close as possible for our evaluation. Finally, superpressure balloon measurements of altitude during the first field campaign of Strateole-2 are used to construct and evaluate power spectra of temperature and vertical velocity (Podglajen et al., 2016b; Haase et al., 2018; Corcos et al., 2021). Strateole-2

balloons were launched from the Seychelles and floated around the Equator in a Lagrangian sense for ~80 days with a goal of determining chemical and momentum fluxes in the tropical tropopause layer. We map fluctuations in altitude at a given time $i$ to those in vertical velocity using the 30-second measurement frequency and to those in temperature assuming dry adiabaticity ($T'_i = -z'_i g/c_p$). A Butterworth filter is used to filter the altitude fluctuations with periods greater than 15 minutes.

## 2.2 Other Observations and Reanalysis

We also assess the ICON model output against several other reanalysis and observational datasets. We use temperature, specific humidity, and cloud ice mass mixing ratio of the ERA-5 reanalysis of the European Centre for Medium-Range Weather Forecasting at 0.25° resolution and hourly frequency (Hersbach et al., 2020). Then we use Aura Level 2 Water Vapor (H2O) Mixing Ratio V005 (ML2H2O) and Temperature V005 (ML2T) products of the Microwave Limb Sounder (MLS) (Waters et al., 2006) with a single overpass transecting our subdomain of interest on 2017-08-08 (Figure 1a, gray). MLS averaging kernels are used in comparing MLS data to in-situ measurements. We also download radiosonde data from the University of Wyoming Atmospheric Soundings database on 2017-08-08 at 00:00 UTC preceding Flight 7 (Sondes 42379, 42492, 42701, 42867, 42809, and 42361; Figure 1a, pink stars). As for the synthetic flight tracks, we calculate vertical profiles of observational differences by binning the in-situ data into the pressure levels of ERA-5, MLS, or the sondes and taking statistics over those binned values.

## 2.3 ICON Simulations

We run six simulations using the ICON model, version 2.6.4, and various couplings of its cloud microphysics and optics (Table 1) (Giorgetta et al., 2018). Both the one- and two-moment microphysics of Doms et al. (2005) and Seifert and Beheng (2006), respectively, are coupled to a hierarchy of three optical schemes in the ecRad radiative transfer scheme. The Fu (1996) and Fu et al. (1998) schemes are the simplest schemes and the default option in ICON. They assume all ice crystals are hexagonal with sizes derived from in-situ measurements. Then the Baran et al. (2016) scheme includes hexagonal ice columns with unit aspect ratio, along with "3-, 5-, 8- and 10-branched hexagonal ice aggregates" with a description of ice crystal surface roughness for shortwave radiative properties. Optical properties depend on temperature, as a proxy for ice crystal habit (which itself is highly temperature dependent). Lastly, Yi et al. (2013) include "droxtals, plates, solid and hollow columns, solid and hollow bullet rosettes, an aggregate of solid columns, and a small/large aggregate of plate" for a total of 9 different habits, again with a description of ice crystal surface roughness. Sepúlveda Araya et al. (2025) document optical parameters and behavior of these schemes in greater detail.

For all possible combinations of microphysics and optics, ICON is run over 4 days from 5 August 2017 12:00 UTC to 9 August 2017 12:00 UTC for an Asian monsoon domain from 55°E to 115°E and from 5°S to 40°N. We use a 24-second timestep, 75 vertical levels, and 2.5-km-equivalent resolution (R2B10 icosahedral grid), so that deep convection is explicit. Shallow convection remains parameterized with the Nordeng (1994) scheme. The Integrated Forecast System (IFS) analysis data of the European Center for Medium-Range Weather Forecast (ECMWF) provides initial and 3-hourly boundary conditions. All trace gas mixing ratios are set to the annual global means from the input4MIPS project. No output is generated until a restart file at 8 August 2017 03:00 UTC; thereafter, both 2D and 3D fields are generated at 10-minute frequency until 8 August 2017 07:00

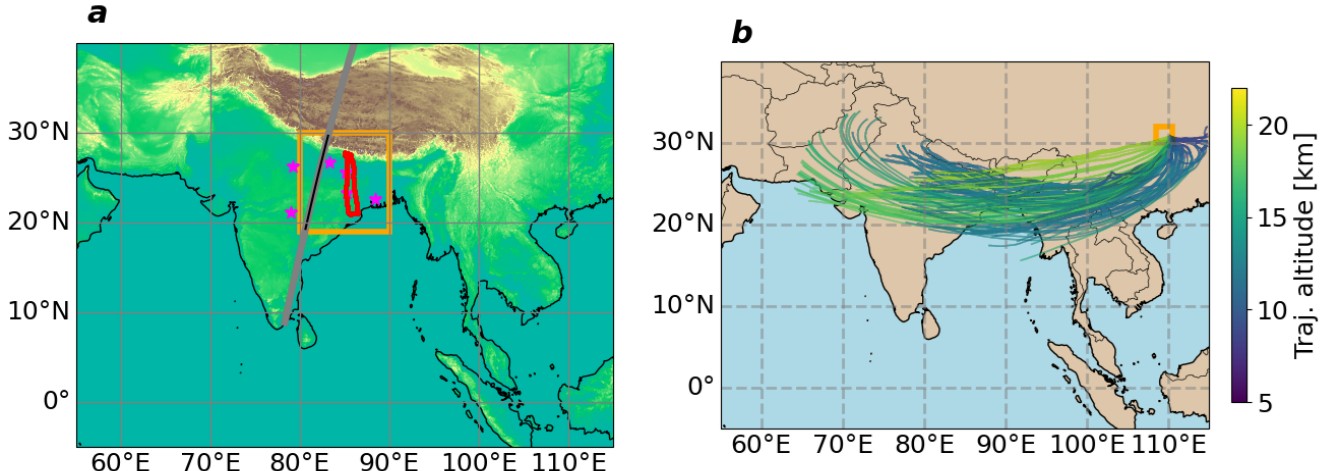

**Figure 1.** Simulation domain over the Asian monsoon region (panel a) with StratoClim Flight 7 track shown in red. Gray: Microwave Limb Sounder swath; pink stars: radiosonde locations; orange box: subdomain around Flight 7, 80°E–90°E and 19°N–30°N. Illustration of online trajectories in ICON, 2.3.0 over our simulation domain colored by trajectory height (panel b). Yellow box: patch over which trajectories initiate.

UTC for a subdomain extending from 19 to 30°N and from 80 to 90°E (Figure 1a). This timing and subdomain correspond to Flight 7 of the StratoClim field campaign (Figure 1a, red).

**Table 1.** ICON simulations; _M denotes microphysics schemes and _O optical ones

| Designation | Simulation setting |
| --- | --- |
| **ICON 2.6.4** | |
| _M | 1 = One-moment microphysics (Doms et al., 2005) |
| | 2 = Two-moment microphysics (Seifert and Beheng, 2006) |
| _O | 0 = Fu (1996) SW and Fu et al. (1998) LW cloud optics |
| | 1 = Baran et al. (2016) cloud optics |
| | 3 = Yi et al. (2013) cloud optics |
| **ICON 2.3.0** | |
| 2M0Ot | Trajectories |
| 2M0Ot_T' | Trajectories + parameterized temperature fluctuations |

### 2.4 ICON Output in Lagrangian Formats

#### 2.4.1 Synthetic Flight Tracks

An ensemble of **synthetic flight tracks** is generated from the 10-minute frequency ICON output to perform model-measurement comparison. We define a domain of $\pm 0.25°$ x $0.25°$ around the StratoClim flight position and randomly sample 20 values with replacement from within this domain for the time and pressure level closest to the aircraft. We continue this random sampling at every time point along the actual flight track to create 20 "flight tracks" from model output. This sampling strategy accounts for possible spatiotemporal offsets between simulated and observed features. We test how different the 20 synthetic flight tracks are relative to values/biases evaluated over a small subdomain around the Flight 7 track (Figure 1a, orange). We calculate vertical profiles of model bias using the synthetic and actual flight tracks, binning the data into the pressure levels of the model and taking statistics over those binned values. Comparisons of vertical profiles focus on the times between 06:20 and 06:48 UTC on 8 August 2017 as in Lee et al. (2019), since the aircraft performs a continuous descent during this period, sampling the full altitudinal range.

#### 2.4.2 Online Trajectories

A trajectory module was also implemented into ICON, version 2.3.0, as detailed in Miltenberger et al. (2020). These online trajectories use an implicit Euler scheme and the resolved wind fields of the simulation to calculate kinematic airmass trajectories. Calculations are performed on the native ICON grid at the physics timestep of the model and interpolation of wind field and trace variables to the trajectory position uses triangular horizontal and linear vertical interpolation. At every half level between 8 and 22 km and at 09:00 UTC on 6 August 2017, we initiate trajectories at all grid cell centers between 30°N and 32°N and between 108.5°E and 110.5°E in the east of our simulation domain (Figure 1b, yellow). This region was the source of overshooting convection that moistened the lower stratosphere, according to the study of Lee et al. (2019). We allow the online trajectories to propagate over 51 hours with the 24-second timestep of the model, as they flow westward toward the region where our observations are located (Figure 1a). We use the temperature and vertical winds along these trajectories to compare to superpressure balloon measurements in a statistical sense. We show results only of a 2M0O setup, denoted 2M0Ot, for these online trajectories (Table 1).

## 3 Results

### 3.1 Evaluating UTLS Temperature

We first visualize the temperature over time of Flight 7 and our synthetic flight tracks (Figure 2). This comparison helps to validate that the simulated values have been correctly extracted. Agreement is reasonable with 65% of the temperature biases within $\pm 2$ K and 82% within $\pm 3$ K; however, 10% of the temperature biases remain larger than $\pm 5$ K. Temperature biases show a strong negative correlation with observed temperature (r = -0.51, $p \ll 10^{-5}$), with the strongest relationships occurring

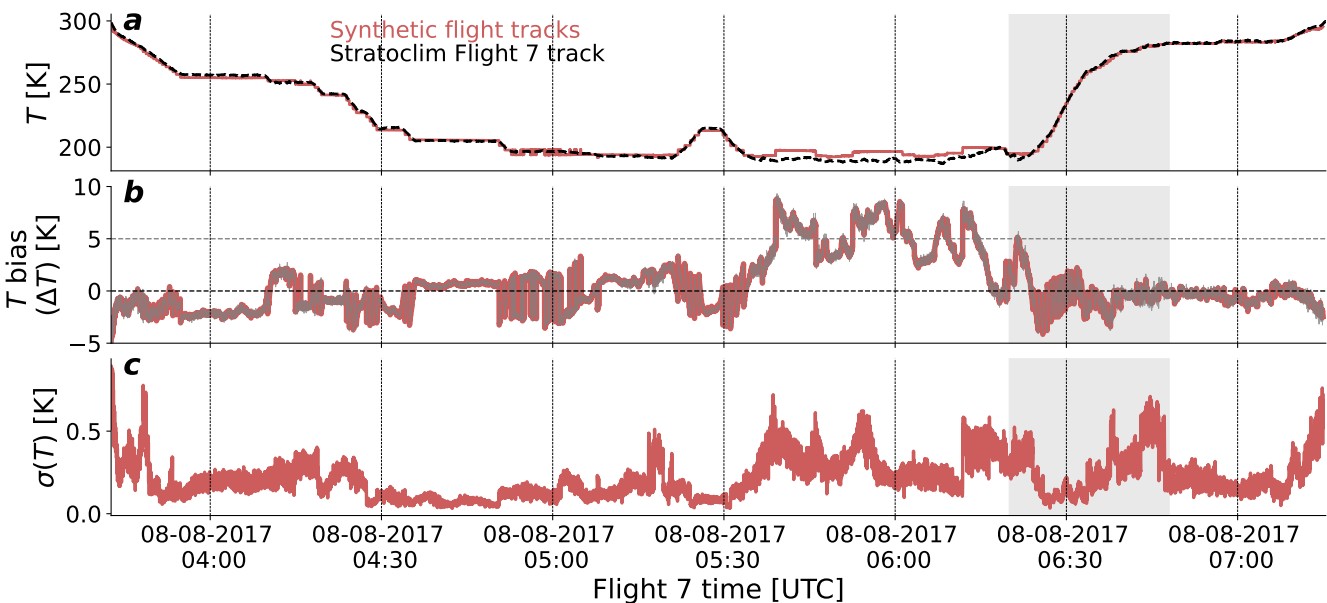

**Figure 2.** Temperature along the StratoClim Flight 7 track and the average temperature across the 20 synthetic flight tracks from the ICON 1M0O simulation (panel ***a***). Temperature biases along the synthetic flight tracks relative to the in-situ measurements (panel ***b***). The temperature bias of the "best trajectory" for which bias is minimized most often is overlaid in gray. Standard deviation in temperature over the 20 synthetic flight tracks (panel ***c***). A period of continuous descent from 06:20–06:48 UTC is shaded in gray for all panels.

below 195 K (r = -0.68, $p \ll 10^{-5}$). All biases greater than ±5 K are associated with observed temperatures colder than 200 K. Along the flight track, these largest biases occur between 05:30 and 06:30 UTC, when the aircraft is flying at the highest altitudes.

To understand spatial persistence of these biases, we examine the "best" synthetic flight track, the one for which the bias is most often minimized (Figure 2b, gray). Again, for this best track, simulated values are warm-biased 85% of the time for temperatures colder than 200 K. As another metric of evaluation robustness, we plot the standard deviation in temperature ($\sigma(T)$) across the synthetic flight tracks (Figure 2c). Seventy five percent of the time, $\sigma(T)$ is less than 25% as large as the bias. $\sigma(T)$ also correlates only weakly with the aircraft descent or ascent, as quantified in $dT/dt$ (r = -0.011, $p < 10^{-2}$; not shown), and with temperature itself (r = 0.13, $p \ll 10^{-5}$). Our temperature evaluation is thus robust across space and the synthetic flight tracks.

The vertical distribution of UTLS temperature, including tropopause position, is crucial for moisture transport into the stratosphere and overshoot occurrence, as mentioned earlier. Vertical profiles of $T$ and $\theta$ have smaller, negative biases at pressures greater than 110 hPa and larger, positives ones at pressures less than 110 hPa (Figure 3a-b). The maximum temperature bias at pressures greater than 110 hPa (+3.5 K) is 2.5 times larger in magnitude than that at pressures less than 110 hPa (-1.3 K). Profiles of temperature bias are qualitatively consistent if we average over a small domain around Flight 7, rather than along

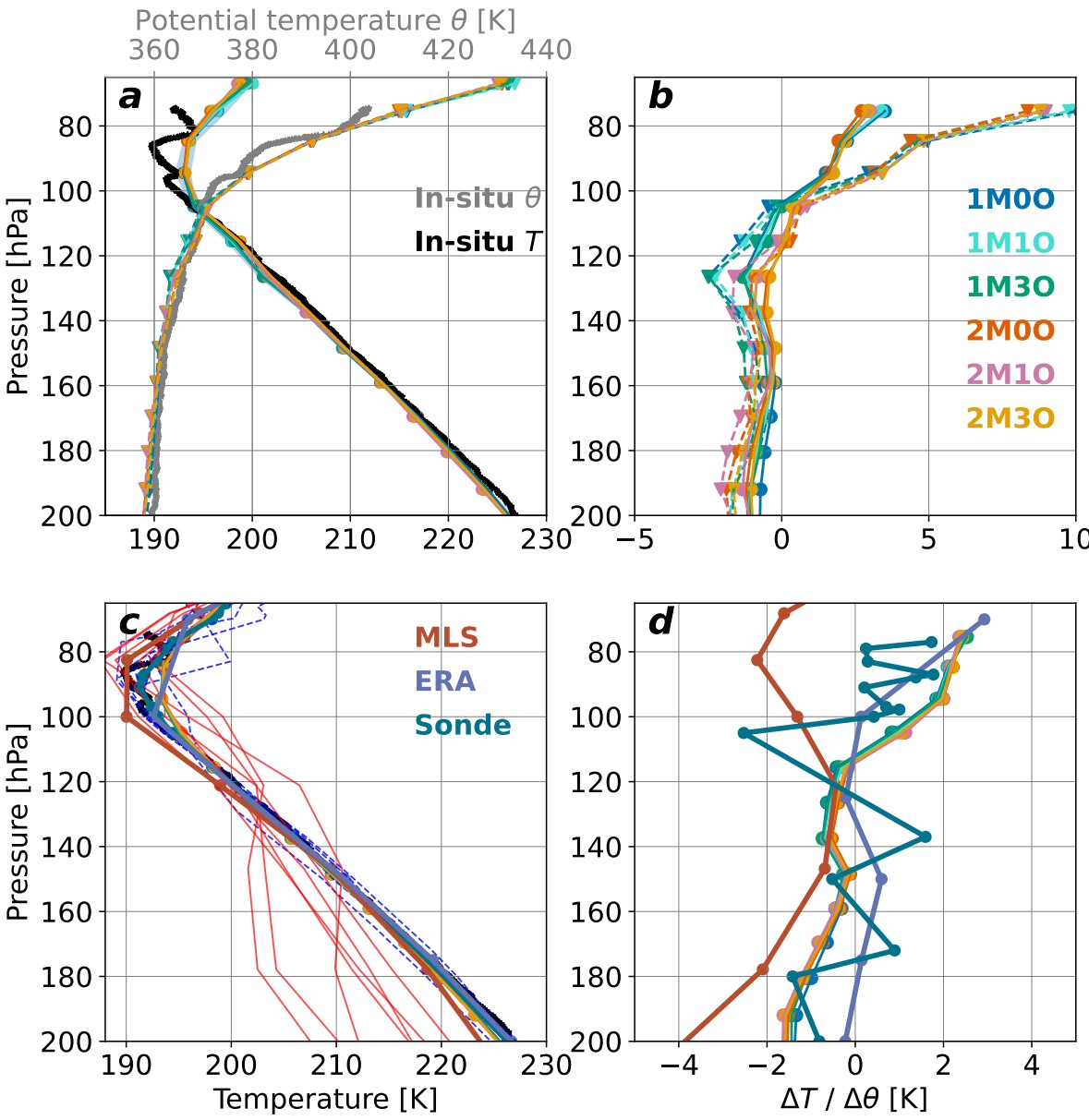

**Figure 3. ICON simulations are too cold in the upper troposphere and too warm in the lower stratosphere.** Comparisons of in-situ temperature $T$ and potential temperature $\theta$ profiles with those from 20 synthetic flight tracks in the 6 ICON simulations for the period of aircraft descent from 06:20-06:48 UTC (panel **a**). Profiles of $T$ and $\theta$ biases relative to in-situ values (panel **b**). For these panels, $T$ is shown in circles and solid lines, and $\theta$ is shown in triangles and dashed lines. Comparisons of in-situ $T$ to those from the Microwave Limb Sounder (MLS) with separate swaths in thin red-orange lines and the closest swath to the Flight 7 track in bold red, the ERA-5 reanalysis in purple, and sondes launched nearby with individual stations in blue dashed lines and their average in solid blue-green (panel **c**). ICON simulation profiles in panel **c** are shown averaged over the entire small subdomain around Flight 7, rather than from synthetic trajectories. Profile of $T$ differences relative to in-situ values for all data in panel c (panel **d**).

the synthetic flight tracks (Figure 3c-d). The vertical differences in temperature are magnified for potential temperature ($\theta$) at low pressures: The max bias in $\theta$ at pressures less than 110 hPa (+10.0 K) is 4 times larger in magnitude than that at pressures greater than 100 hPa (-2.5 K). Inter-simulation differences are modest with temperature biases varying by at most 0.8 K, and we discuss these more thoroughly in Section 3.2.

Lastly, we examine robustness of the temperature biases to other observational baselines (Figure 3c-d). The ERA-5 reanalysis agrees well with the in-situ measured temperatures except at the highest vertical level in the stratosphere, where it predicts a warmer temperature in line with ICON. The sondes near Flight 7 also tend to register warmer temperatures than the in-situ measurements at pressures lower than 100 hPa; for pressures greater than 100 hPa, the sonde-in-situ differences are noisy. The Microwave Limb Sounder (MLS) records 8 swaths in the small subdomain around the Flight 7 track, and several of these show much colder upper tropospheric temperatures than any of the other datasets (Figure 3c, thin red lines). Even the swath geographically closest to the flight track is consistently colder than the in-situ measurements after application of MLS averaging kernels. If we had used MLS as our baseline, we would then report a much larger stratospheric warm bias and no upper tropospheric cold bias. This discrepancy points to the difficulty of establishing an observational truth in evaluation of UTLS structure.

## 3.2 Evaluating UTLS Moisture

We next consider moisture structure with a time series of specific humidity ($q_v$) from the synthetic and actual flight tracks (Figure 4). The FLASH-A instrument only measures at pressures below 250 hPa for about 2 hours from 04:35 to 06:30 UTC to avoid contamination by the abundant moisture at lower altitudes. Again, the synthetic flight tracks generally match the behavior of the in-situ data. A relative bias makes more sense than an absolute one in this case, as $q_v$ spans a huge range of values. Normalizing by the observation, 24% of time points fall within $\pm10\%$ of the observed $q_v$, 58% fall within $\pm20\%$, and 99% fall within $\pm50\%$, relative to a 6% measurement uncertainty for FLASH-A (Figure 4b).

Dry biases are two times more frequent than moist ones. The dry biases correlate negatively with $q_v$ ($r$ = -0.34, $p \ll 10^{-5}$) and the moist biases correlate positively ($r$ = 0.22, $p \ll 10^{-5}$), meaning that the model makes the drier locations too dry and the moister ones too moist. The worst dry biases consistently occur for $q_v$ values from 5 to 10 ppmv, reinforcing this point. The $q_v$ biases do not correlate strongly with observed temperature or aircraft ascent/descent ($dT/dt$) but are similar to those in temperature in that their general behavior is not track-specific. Examining the $q_v$ biases from the "best-performing track", we find more inter-track differences for $q_v$ than for $T$ but that these remain small relative to the magnitude of the bias itself (Figure 4b). The standard deviation in $q_v$ across the trajectories reinforces this point (Figure 4c): $\sigma(q_v)$ is almost always less than 10%, while only a quarter of the $q_v$ biases are within $\pm10\%$, as noted above.

In vertical profiles of $q_v$, ICON captures the observed moisture fairly well at pressures lower than 160 hPa, given its limited vertical resolution (Figure 5a-b); however, it does miss anomalously high $q_v$ around 80 hPa, a "hydration patch" traced back to convective overshoots over the Sichuan basin 36 hours prior by Lee et al. (2019). $q_v$ increases almost twofold over 10 hPa in the observations (from 3.3 to 5.8 ppmv between 87 and 78 hPa), whereas it increases only by 20% over 20 hPa in the model (from 4.1 to 5.2 ppmv between 94.5 and 75.2 hPa). In contrast, ICON overestimates moisture in two upper-tropospheric layers from

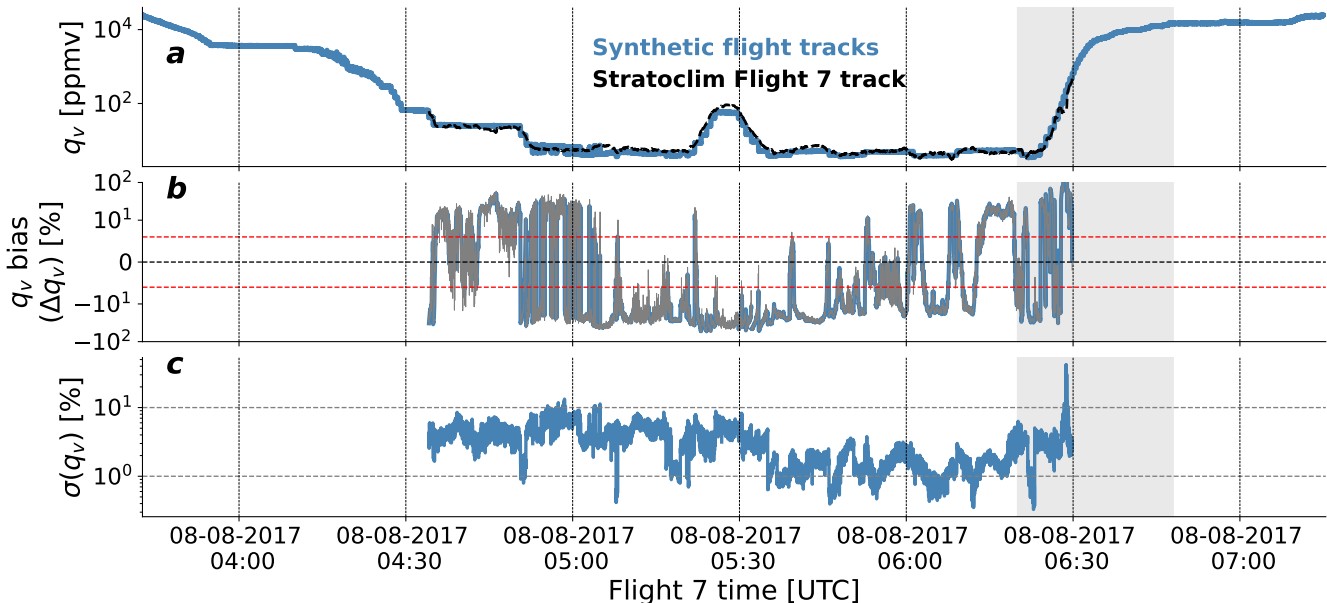

**Figure 4.** Specific humidity along the StratoClim Flight 7 track and averaged from the ICON 1M0O simulation synthetic flight tracks over time (panel *a*). Specific humidity biases along the synthetic flight tracks relative to the in-situ measurements (panel *b*). The specific humidity bias of the "best trajectory" for which bias is minimized most often is overlaid in gray and measurement uncertainty of 6% is shown in red. Standard deviation in specific humidity over the 20 synthetic flight tracks (panel *c*). A period of continuous descent from 06:20–06:48 UTC is shaded in gray for all panels.

140-120 hPa and from 200-160 hPa. As for temperature, these $q_v$ biases remain if we average over the Flight 7 subdomain, rather than along the synthetic flight tracks (Figure 5c-d). The ERA-5 reanalysis shows similar $q_v$ behavior to ICON, reflecting that the hydration patch aloft and dryness below are both quite localized vertically (Figure 5c-d). The kernel-averaged MLS profiles somewhat ~20% more moisture below 140 hPa and 20% less above. These inter-measurement differences again point to the difficulty not only in simulating UTLS structure accurately but also in observing it accurately.

The vertical profiles suggest covariance between the biases in temperature and specific humidity, and we visualize this more clearly in joint distributions, constructed for all of the flight track above 14 km, not only the descent period from 06:20-06:48 UTC (Figure 6). UTLS biases are most often too warm and too dry and rarely too cold and too moist. We understand this result to mean that the behavior in ICON above 120 hPa—with overestimated $T$ and underestimated $q_v$—is the most widespread. The distributions also show that the two-moment simulations are more sensitive to the optics scheme used, as covariance of biases shifts more for these runs than the one-moment ones. We highlight this point again in Section 4.

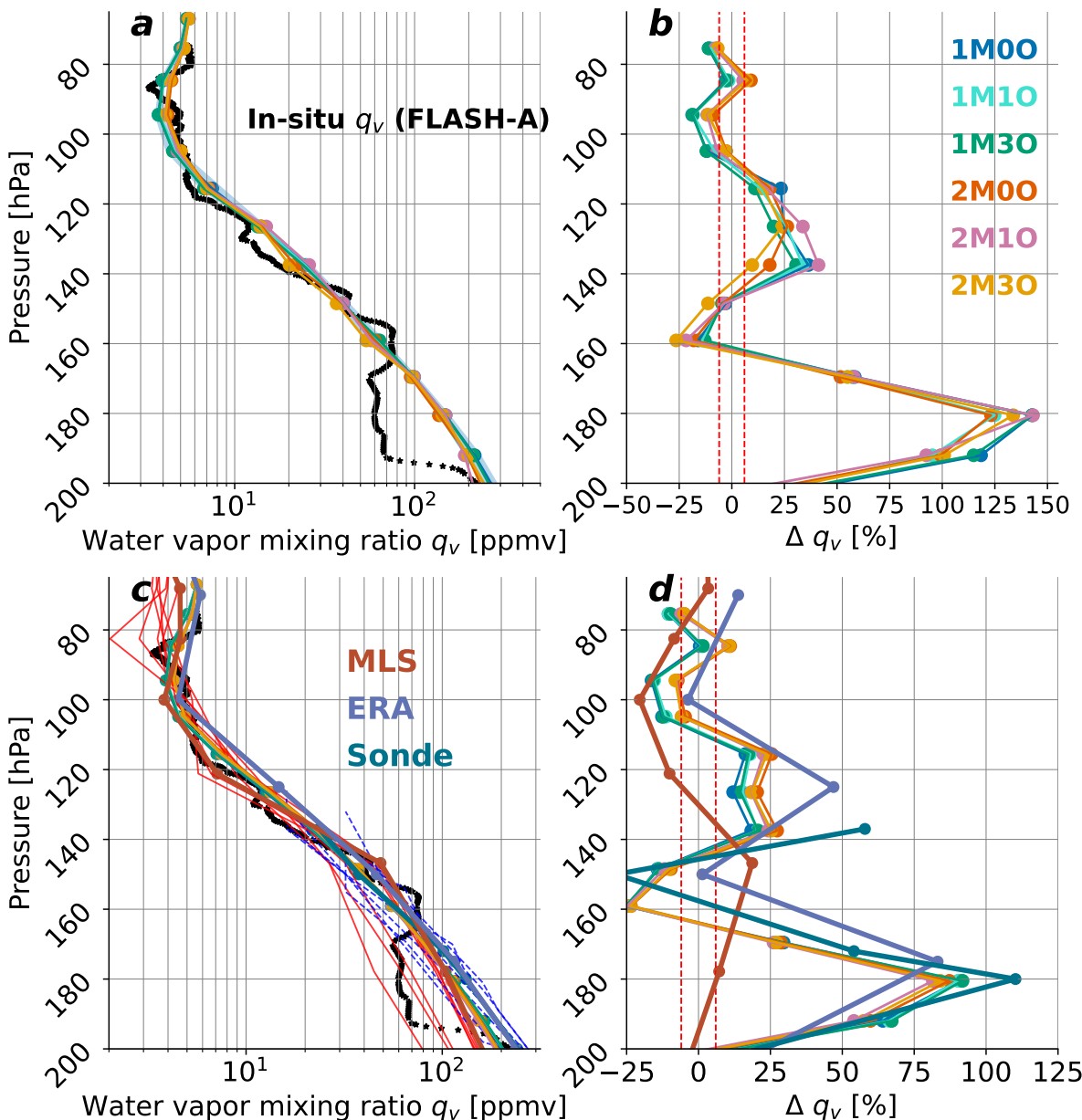

**Figure 5. ICON struggles to capture localized hydration aloft and dryness below.** Comparisons of in-situ specific humidity $q_v$ profiles with those from the ICON synthetic flight tracks for the period of aircraft descent from 06:20-06:48 UTC (panel **a**). Profiles of $q_v$ biases relative to in-situ values, measurement uncertainty in dashed red (panel **b**). Comparisons of in-situ $q_v$ to those from the Microwave Limb Sounder (MLS) with separate swaths in thin red-orange lines and the closest swath to the Flight 7 track in bold red, the ERA-5 reanalysis in purple, and sondes launched nearby with individual stations in blue dashed lines and their average in solid blue-green (panel **c**). ICON simulation profiles in panel **c** are shown averaged over the entire small subdomain around Flight 7, rather than from synthetic trajectories. Profile of $q_v$ differences relative to in-situ values for all data in panel c, measurement uncertainty in dashed red (panel **d**).

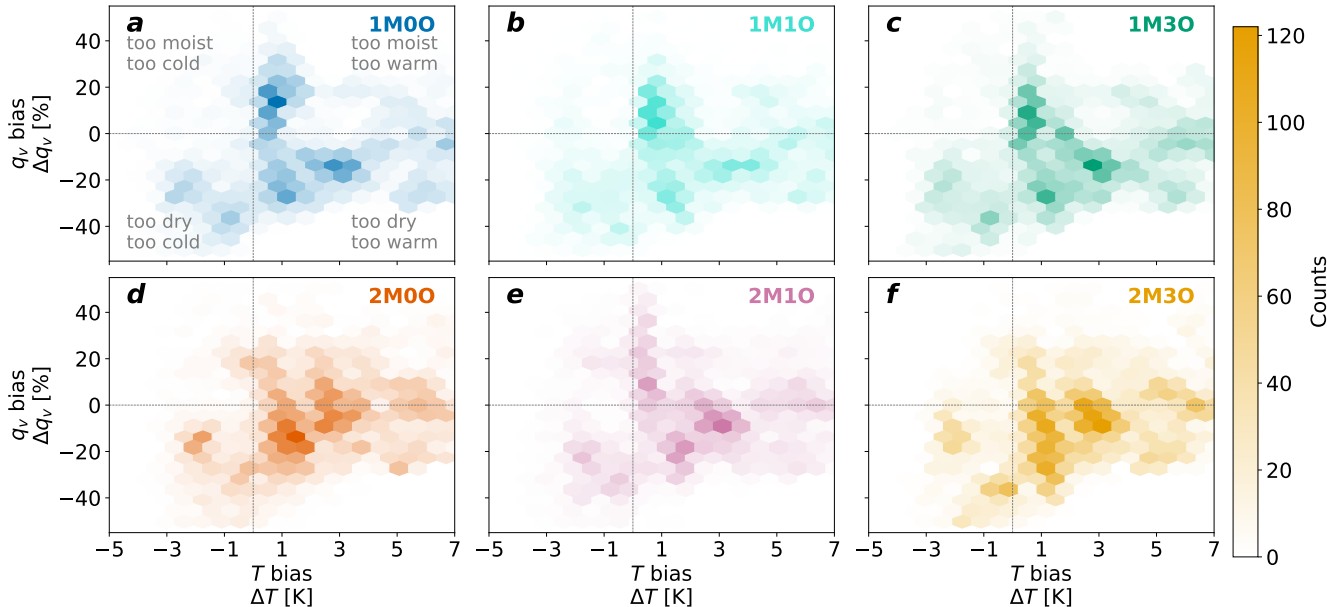

**Figure 6. Too-warm and too-dry biases are most common for the upper troposphere-lower stratosphere in ICON.** Joint distributions of the absolute bias in temperature ($\Delta T$) and the relative bias in specific humidity ($\Delta q_v$) for all of the in-situ and synthetic flight tracks above 14 km. The six panels correspond to the six simulations listed in Table 1. Four quadrants are denoted in panel *a*.

## 3.3 Evaluating UTLS Vertical Velocities

To evaluate the UTLS ascent rates in ICON, we turn to observations of the ATTREX and POSIDON campaigns, as $w$ was not measured during StratoClim (see Section 2.1). The evaluations can only be statistical then, with probability distributions of updraft and downdraft velocities between 200 and 70 hPa (Figure 7a-b). The mean observed updraft speed is 0.40 m s$^{-1}$ from POSIDON and 0.20 m s$^{-1}$ from ATTREX with a standard deviation in these speeds of 0.31 and 0.17 m s$^{-1}$, respectively. POSIDON sampled close to deep convection more often, yielding its stronger and more variable speeds. Simulated updraft velocities are slower with a mean value of 0.077 m s$^{-1}$ and have less spread with a standard deviation of 0.11 m s$^{-1}$. The same is true for downdrafts: While ATTREX and POSIDON record mean downdraft speeds of -0.20 and -0.40 m s$^{-1}$ and standard deviations of 0.45 and 0.16 m s$^{-1}$, downdrafts in the ICON simulations have an average speed of -0.073 m s$^{-1}$ and spread of 0.078 m s$^{-1}$. Inter-simulation differences in the updraft statistics are limited with the one-moment simulations predicting updraft speeds 12% faster and downdraft speeds 6.5% faster.

Along with convective ascent, upper-tropospheric gravity waves generate temperature and moisture fluctuations that influence ice formation (e.g., Potter and Holton, 1995; Podglajen and Plougonven, 2018). The Lagrangian format of the online ICON trajectories allows us to assess the simulated gravity wave spectrum in the model (Figure 1b). As noted in Section 2.4, these online trajectories come from a Flight 7 simulation in ICON version 2.3.0, whereas other evaluations use an identical setup in ICON version 2.6.4. The same dynamical core, two-moment microphysics, Fu optics, and initial and boundary con-

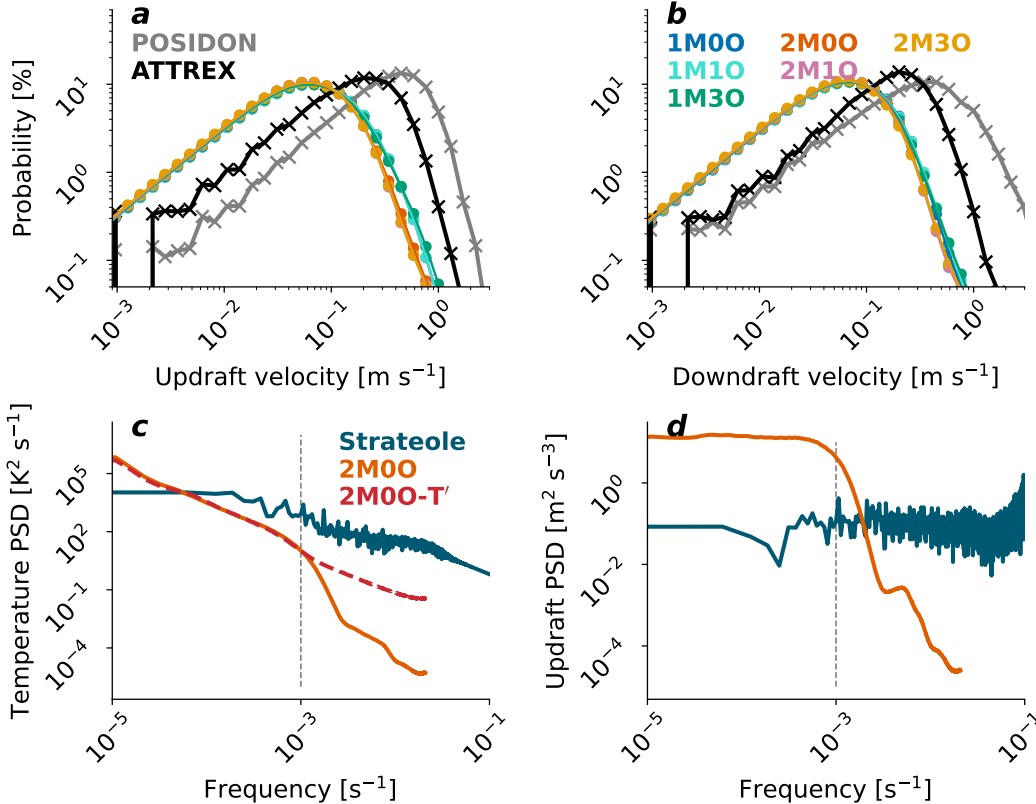

**Figure 7. Strong vertical motions and high-frequency temperature and vertical velocity fluctuations are underestimated in the ICON simulation.** Probability distributions of updraft (panel **a**) and downdraft (panel **b**) velocities between 200 and 70 hPa from the six ICON simulations (colored traces), relative to those of the POSIDON (gray) and ATTREX (black) field campaigns in the west Pacific tropical tropopause. Power spectra for temperature for the ICON 2M0O trajectories, both with (magenta) and without (orange) temperature fluctuations, and relative to the Strateole-2 superpressure balloon measurements (panel **c**). Power spectra for updraft velocities in both ICON 2M0O and Strateole-2 data (panel **d**). The dashed gray lines in panels c and d indicate the upper bound in resolvable frequencies at 2.5-km grid spacing, assuming a gravity wave speed $\sim \mathcal{O}(10 \text{ m s}^{-1})$.

ditions are used across these two versions, and the online trajectories are not yet implemented in the more recent version. We expect that qualitative and approximate quantitative results from the version 2.3.0 trajectories would hold also in version 2.6.4.

With a horizontal resolution of $\sim$2 km, we only expect to resolve waves of 8-km wavelength or larger (Figure 7, gray dashed line). Indeed, in power spectra of temperature and vertical velocity fluctuations along the ICON trajectories, we see dramatic underestimations of high-frequency power despite their 24-second output frequency (Figure 7c-d). Relative to the power spectra of Strateole-2 measurements, ICON underestimates the power for frequencies on the order of $10^{-2}$ s$^{-1}$ by almost four orders of magnitude in both $T$, reflective of lower-frequency gravity waves, and $w$, reflective of higher-frequency gravity waves. Similar magnitude underestimations have been found between spectra of kinetic and potential energy from

235

reanalysis versus balloon measurements (Podglajen et al., 2020), as well as between spectra simulated in the COSMO-Model at 2-km resolution versus aircraft observations (Kienast-Sjögren et al., 2015). More recently, the review of Achatz et al. (2024) has also discussed the inability of even kilometer-scale simulations to fully resolve the gravity wave spectrum due to omission of individual convective cells and interactions of gravity waves with turbulence.

Given this large underestimate of high-frequency motions in the ICON simulations, we activate a parameterization of temperature fluctuations, derived from measurements of another superpressure balloon campaign, PreConcordiasi (Podglajen et al., 2016a):

$$z'_{i+1} = z'_i + W \cdot dt \tag{1}$$
$$T'_{i+1} = -\frac{g}{c_p} z'_{i+1} \tag{2}$$

where the altitudinal displacement from one time step $i$ to the next $i+1$ is determined by a white noise process $W$ associated with vertical velocity fluctuations with an observationally derived variance of 0.16 m s$^{-1}$. A high-pass filter removes any fluctuations with frequency lower than 2 days, and altitudinal perturbations are mapped to temperature ones assuming dry adiabaticity. Inclusion of this parameterized temperature variability lifts the high-frequency tail ($f > 10^{-3}$ s$^{-1}$) of the temperature power spectrum by about two orders of magnitude from $10^{-4}$ K$^2$ s$^{-1}$ up to $10^{-2}$ K$^2$ s$^{-1}$, so that the decrease in power with frequency aligns with the -2-slope of the Strateole-2 measurements (Figure 7c). From this last analysis, we conclude that balloon-informed parameterizations of gravity wave activity could improve the UTLS realism of even kilometer-scale simulations.

### 3.4 Evaluating UTLS Ice Clouds

We lastly consider ice cloud formation along the real and synthetic flight tracks (Figure 8). We mask instances of negligible $q_i$ (defined as $q_i < 0.001$ ppmv), so that the $q_i$ time series are discontinuous but with a large amount of time spent sampling within cirrus nonetheless: 50.3% in the simulation tracks and 50.1% in the real one. Although this in-cloud duration agrees well, the timing and magnitude of in-cloud $q_i$ differ. Ice clouds tend to be predicted but not observed when the aircraft measures warmer temperatures and higher specific humidities and the model underestimates $T$ and overestimates $q_v$. The model cold bias is the strongest driver of these overestimates in ice cloud occurrence, as quantified by a t-statistic between distributions for the full $q_i$ series and masked for periods of $q_i$ overestimate ($t_{\Delta T}$ = -43.7, $p << 10^{-5}$). When ice clouds are observed but not predicted, conditions are just the opposite—temperatures are colder, specific humidities are lower, and the model exhibits overestimates in $T$ and underestimates in $q_v$. The model warm bias is again the strongest driver of the underestimates in ice cloud occurrence ($t_{\Delta T}$ = 37.5, $p << 10^{-5}$).

Vertical profiles of $q_i$ corroborate this picture in which ice cloud is underestimated for warm-dry-biased pressure levels above 100 hPa and overestimated for cold-moist-biased pressure levels below 160 hPa (Figure 9). The too-warm and too-dry conditions in the lower stratosphere are consistent with the lack of ice clouds there and suggest that the overshooting convection at the root of moisture and ice aloft during Flight 7 could be underrepresented in ICON (Lee et al., 2019). The aircraft also samples almost no ice clouds at pressures greater than 160 hPa, while the model simulations predict $q_i$ from 10-100 ppmv there.

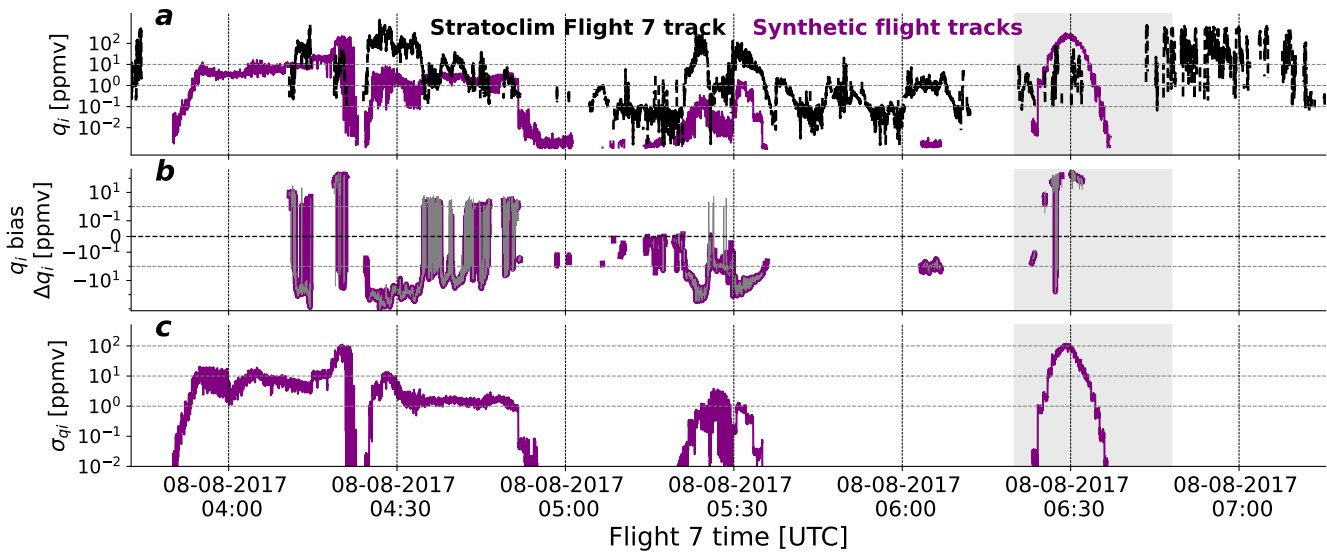

**Figure 8.** Ice mass mixing ratio along the StratoClim Flight 7 track and averaged from the ICON 1M0O simulation synthetic flight tracks over time (panel **a**). Ice mass mixing ratio biases along the synthetic flight tracks relative to the in-situ measurements (panel **b**). The ice mass mixing ratio bias of the "best trajectory" for which bias is minimized most often is overlaid in gray. Standard deviation in ice mass mixing ratio over the 20 synthetic flight tracks (panel **c**). A period of continuous descent from 06:20–06:48 UTC is shaded in gray for all panels.

High-frequency gravity waves generally act to amplify, rather than quench, ice crystal formation near the tropical tropopause (Jensen et al., 2016), so the underestimation of strong updrafts and gravity wave activity likely suppress cloud ice aloft.

The spread in $q_i$ for these altitudes is the largest inter-track difference. Qualitative behavior is the same across runs but
a quantitative hierarchy emerges in which all the one-moment simulations predict larger $q_i$ than the two-moment ones. The simulations that employ the Yi et al. (2013) ice optical scheme with hexagonal ice aggregates (∗M3O runs) bound the range of $q_i$. Interestingly then, our synthetic flight tracks can be approximately interpreted as *Lagrangian piggybacking* in which thermodynamic inputs are held constant while cloud physics and optics change (Sullivan et al., 2022). Our findings echo those of Sullivan et al. (2022) in which altering ice microphysics for fixed (thermo)dynamic inputs generates a 5-fold difference in
upper-tropospheric $q_i$.

We close by illustrating the localization of ice clouds in a few ways. First, we get $q_i$ values about 5-fold smaller if we take the median across the synthetic flight tracks rather than the mean (Figure 9c, triangles). Many grid cells sampled from the model output have much lower $q_i$ then and a few instances of strong ice formation pull the mean $q_i$ higher. Then, if we take the mean $q_i$ over a subdomain around the Flight 7 track rather than along our synthetic flight tracks, $q_i$ can be twice as small below 160
285    hPa (Figure 9c, squares). In other words, the $q_i$ evaluation is more sensitive to the sampling method for evaluation than either $T$ or $q_v$. Finally, in the time series of $q_i$, the standard deviation in ice mass mixing ratio ($\sigma_{qi}$) across the synthetic flight tracks can be on the same order as $q_i$ itself; in contrast, $\sigma_T$ only reaches ∼0.5% of temperature values.

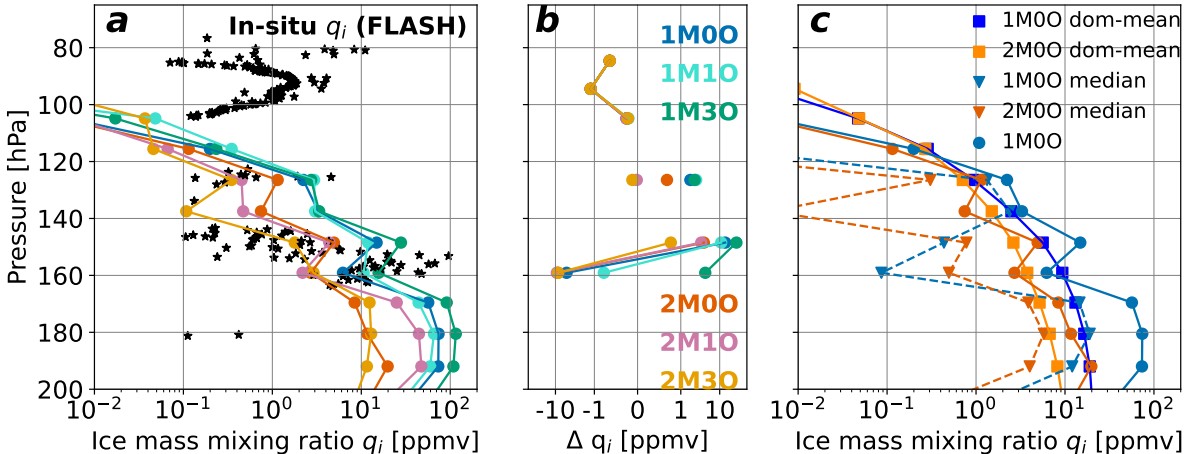

**Figure 9. Ice cloud occurrence is overestimated in the upper troposphere and underestimated in the lower stratosphere.** Comparisons of in-situ ice mass mixing ratio $q_i$ profiles with those from the ICON synthetic flight tracks for the period of aircraft descent from 06:20-06:48 UTC (panel **a**). Profiles of $q_i$ biases relative to in-situ values (panel **b**). Mean $q_i$ profiles for the 1M0O and 2M0O simulations in circles and solid lines and median $q_i$ profiles in triangles and dashed lines (panel **c**). Profiles taken over the whole subdomain around the Flight 7 track (Figure 1a, orange) are shown in squares and solid lines.

## 4  Discussions

We have evaluated tropical upper troposphere-lower stratosphere (UTLS) structure in the ICON storm-resolving model in several ways. We use unique high-altitude measurements of the StratoClim field campaign to assess simulated temperature, moisture, and ice cloud fields, both over time and binned vertically. We also use both flight campaign data of ATTREX and POSIDON and superpressure balloon data of Strateole-2 to assess simulated convective updrafts/downdrafts and gravity wave activity above 14 km in ICON. In this evaluation, a picture emerges in which the upper troposphere is slightly too cold and occasionally too moist, leading to overestimates of ice cloud formation there relative to in-situ observations. Meanwhile, the lower stratosphere is too warm, too dry, and too quiescent, suppressing ice cloud formation there relative to in-situ observations. Of these biases, the overestimates of lower stratospheric temperature are the most robust.

We have tested robustness of the evaluation in several ways. First, we account for spatial offset using synthetic flight tracks from model output, rather than simply averaging the output over a domain around the actual flight track. Ice mass mixing ratio can vary by a factor of 2–5 between these methods, but otherwise we do not gain much from the Lagrangian evaluation, as the biases behave similarly across these two methods. Second, we explore how different biases would be if we used satellite, sonde, or reanalysis data as our observational baseline. Temperature and especially moisture biases would be quite different when using these products. For example, the Microwave Limb Sounder would have reported much larger stratospheric warm biases and no stratospheric dry bias. These discrepancies point to another layer of complication in accurately capturing tropical UTLS structure, as well as the value of high-altitude in-situ measurements.

Finally, we look at different representations of ice cloud microphysics and optics. The stratospheric warm and dry biases are somewhat larger in the one-moment microphysics run, but the largest inter-scheme differences by far are in ice mass mixing ratio. We make two final points about model couplings. Joint distributions of temperature and moisture biases indicate that the two-moment runs are more sensitive to the coupled optics. With a freely evolving ice crystal number, rather than one evaluated from $q_i$, the two-moment scheme "explores" more of the parameter space of optical properties. In addition, Lee et al. (2019) were able to more successfully capture the TTL temperatures and moisture from StratoClim Flight 7 with the same 2.5-km equivalent grid spacing as in our simulations but almost twice as many vertical levels. Their relative success emphasizes the importance of refining vertical grid spacing to reproduce UTLS properties. Microphysical and optical sensitivities may also emerge more strongly with refined vertical resolution, and as Achatz et al. (2024) point out, vertical resolution has a large influence on more reliable representation of gravity wave activity, which feeds back upon the representation of ice clouds.

One mechanism consistent with the reported biases is that convective overshoots are too weak or too infrequent. If the Sichuan Basin convection causing the elevated moisture and ice layers observed in Flight 7 were too weak (Lee et al., 2019), then the lower stratosphere would be left too dry, and lack of stratospheric moisture would cause localized warming. The absence of injected ice would also contribute to the negative biases in $q_i$ downstream. Given that our synthetic flight track methodology allows for some spatial offset, it is unlikely that incorrect westward advection of sufficient moisture and ice is responsible. A second mechanism for the biases could be overly rapid dissipation of upper-tropospheric ice clouds. Shorter-lived ice clouds would produce less longwave cooling aloft and less longwave heating below in line with the temperature biases. Too-quick ice growth would also dehydrate too much aloft and hydrate too much below, as the crystals sedimented and then sublimated. The one- and two-moment differences align with this idea. Updraft speeds and ice mass mixing ratio below 160 hPa are systematically larger for the one-moment run. These conditions suggest more rapid ice formation and correspond to the worse warm and dry biases in the lower stratosphere.

Ice crystal complexity is determinative for this latter mechanism. How large, elongated, and oriented the ice crystals are will determine how long they stay aloft in the cloud. Ice cloud microproperties are underdetermined by current remote sensing platforms measuring radiance only. As a result, two-moment microphysics schemes cannot be fully constrained. Spaceborne polarimetry may be one avenue toward better constraint of upper-level cloud schemes and hence toward better representation of UTLS structure. For example, in the microwave band, polarization of ice crystal scattering is particularly useful to infer ice water path, crystal size, and diurnal cycles for the optically thin cirrus that would form in the UTLS (Gong and Wu, 2017; Gong et al., 2018). Recent launches from the Columbia Scientific Balloon Facility in New Mexico and the Esrange Space Center in Sweden prove viability of thermal polarimetry as another path toward additional observational constraints for ice clouds (Shanks et al., 2024; John et al., 2025). These exciting new observations may be one path toward more reliable representation of the complicated and influential UTLS region.

*Code and data availability.* Code to reproduce analysis and figure generation is available in an archived git repository at https://doi.org/10. 5281/zenodo.17252590. All postprocessed data to reproduce figures have been archived at https://zenodo.org/records/17211372. Microwave

Limb Sounder data can be downloaded from NASA's Earthdata portal https://search.earthdata.nasa.gov/search/, and MLS averaging kernels are available at https://mls.jpl.nasa.gov/eos-aura-mls/data.php. ERA-5 data is publicly available at https://cds.climate.copernicus.eu/datasets/reanalysis-era5-pressure-levels. Sonde data can be downloaded from the University of Wyoming archive at https://weather.uwyo.edu/upperair/sounding.shtml. A POSIDON overview is provided at https://espo.nasa.gov/posidon.

*Author contributions.* SCS and AV conceptualized the study with support of CR and MK. AM facilitated implementation of online trajectories in ICON version 2.3.0. SCS completed ICON version 2.3.0 runs, and EISA completed ICON version 2.6.4 runs. CR and MK assisted in handling of in-situ and satellite datasets. AV and MKK supported funding acquisition. SCS wrote the manuscript and generated figures. All authors provided feedback on the manuscript.

*Competing interests.* Some authors are members of the editorial board of the journal ACP.

*Acknowledgements.* SCS acknowledges financial support of an Ida Pfeiffer Professorship from the University of Vienna and from the NASA Earth Science Technology Office through their Instrument Incubator Program, Grant No. 80NSSC25K7306: CHanneled Infrared Polarimeter (CHIRP). EISA and MKK have also been supported by NASA IIP-23 Project CHIRP. We acknowledge compute time from the German Climate Supercomputing Center (DKRZ) under allocation bb1430, as well as a Research Licence of the ECMWF permitting download of ERA-5 reanalysis fields. Thanks go to Eric Jensen for preparing the vertical velocities from ATTREX and POSIDON and to Aurélien Podglajen for the Strateole-2 data and advice on power spectra to characterize gravity waves. The StratoClim field campaign was funded by the European Commission's Seventh Framework Programme (FP7/2007-2013) under grant agreement no. 603557.

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
