# Peer review of "Evaluation of upper-tropospheric lower-stratospheric properties over the Asian monsoon region in a storm-resolving model"

_EGUsphere, 2025_

## Referee Comment (RC2)

**Review report for EGUSPHERE-2025-4981**

**Title:** Evaluation of upper-tropospheric lower-stratospheric properties over the Asian monsoon region in a storm-resolving model.

**Authors:** Sylvia C. Sullivan, Aiko Voigt, Edgardo Sepúlveda Araya, Silvia Bucci, Annette Miltenberger, Meredith K. Kupinski, Christian Rolf, and Martina Krämer

**Recommendation:** Publish after minor revisions

**General Comments**

This manuscript presents a comprehensive evaluation of the ICON model at storm-resolving resolution (2.5 km) over the Asian Monsoon region. By using unique high-altitude in-situ measurements from the StratoClim campaign, along with additional datasets (ATTREX, POSIDON, Strateole-2), the authors provide a valuable assessment of the model's performance in the UTLS region. The study successfully links thermodynamic biases (cold/moist UT, warm/dry LS) to microphysical outcomes (ice cloud placement) and dynamical deficiencies (weak vertical velocities and gravity wave activity). The identification of mechanisms, specifically the underrepresentation of convective overshooting and the potential role of gravity waves, adds significant scientific value.

The manuscript is well-written and logically structured. However, I have noted a few minor typos and instances where figure captions are somewhat unclear or incomplete regarding legend definitions. I recommend publication after minor revisions to address the specific points listed below.

**Specifc Comments**

1. Page 1, line 10: The text mentions "frequencies greater than $10^3$ s$^{-1}$". This corresponds to 1000 Hz (acoustic range), which is physically impossible for atmospheric gravity waves. Section 3.3 correctly identifies the range as f > $10^{-3}$ s$^{-1}$. Please correct the exponent in the abstract to $10^{-3}$ s$^{-1}$.

2. Page 3, line 68: The text mentions "ICON model, version 2.6.4". Please provide the citation or reference for this model version.

3. Page 3, line 86: There is a discrepancy on the latitude range of the subdomain. The text (Page 3, Line 86) states the subdomain extends from "19 to 30°N", whereas the caption of Figure 1 defines it as "20°N–30°N". Please check and ensure consistency between the text and the figure.

4. Figure 1 caption: The text describes the trajectory initiation area as a "gold" box, while the Figure 1 caption refers to it as a "yellow box". Please use consistent wording to avoid confusion.

5. Page 5, line 110: The text refers to the simulation setup as "2M0O", but Table 1 lists the designation as "2M0Ot". Please correct the text to match the designation in the table.

6. Figure3: The caption needs revision to match the panels. Specifically in Panel c:

   (a). It mentions "thin red lines" (which appear orange) but fails to define the solid thick red line (most likely the MLS average).

   (b). The color for the ERA-5 reanalysis (purple) is not specified, whereas other datasets are explicitly described. For consistency, this should be added.
   Please change the caption for Figure 5 as well.

7. Page 9, line 170: Please change "pressure" to "pressures".

8. Page 14, line 253: There is a missing period (".") at the end of the sentence.

9. Page 14, line 269: The text says the aircraft samples "no ice clouds" at pressures greater than 160 hPa. However, Figure 9a shows a few black stars (in-situ data) in this area, which means some ice was present. Please rephrase this to be more precise. For example, use "samples negligible ice clouds" or "samples almost no ice clouds."

10. Figure 9: Please add a legend for panel (c) so the reader can understand what the lines represent.

11. Page 18, line 339 & 340 &346: The initials used for the authors are not consistent. In "Author Contributions," they are written as "ESA" and "MK," but in "Acknowledgements," they are written as "EISA" and "MKK." Please use the same format in both sections.

---

## Author Comment (AC1)

**Reviewer 1 Comments**

This work compares the storm-resolving ICON model, configured with different cloud microphysics and radiation schemes, against several field-campaign observations over the upper-troposphere–lower-stratosphere (UTLS) Asian monsoon region, focusing on temperature, moisture, and cloud ice. The Asian monsoon UTLS is a key region influenced by deep convection and remains challenging for most models. Using a state-of-the-art storm-resolving model to investigate these processes is highly valuable. This work integrates a rich set of model configurations and observational data sources. The analysis is convincing, and the presentation is clear. I did not identify any major issues, only minor consistency points, so I recommend acceptance after a minor revision.

We appreciate the reviewer's time and effort in reviewing our manuscript and for their feedback on our work. Responses are in blue below.

1. Line 31: "Merlis et al., …" I recommend separating this sentence into two, with one describing the overall CMIP6 results and another describing X-SHiELD.
   We have adjusted this wording in the introduction for clarity.

2. Line 67: Please add a citation for the ICON model version 2.6.4.
   We have added a reference to Giorgetta et al. *J. Adv. Model Earth Sys.* (2018) here (https://agupubs.onlinelibrary.wiley.com/doi/full/10.1029/2017MS001242). This article focuses on the default model physics for the atmospheric component of ICON, which should be more relevant to understanding our results than other references focused on the ICON dynamical core or grid refinement.

3. Line 106: "Figure 1b, gold" should be corrected to "yellow," as stated in the figure caption.
   Thank you for catching this. We have corrected it.

4. Section 2.3: I recommend switching the order and moving this section before the introduction of the model experiments, since the experiments are designed to reproduce these field-campaign measurements.
   Thank you for this suggestion. We have moved the subsections about the observations prior to those about the ICON simulations.

5. Section 2.3: Consider also marking the locations of POSIDON and ATTREX in Figure 1.
   Both POSIDON and ATTREX flights were based out of Guam, so their locations unfortunately will not fit on the map shown in Figure 1. We add to the text that these campaigns were both based out of Guam.

6. Figure 2a: I do not think there is a clear reason to invert the y-axis, and doing so could be misleading.

The rationale for an inverted y-axis was that the trace replicated the flight altitude, with low values indicating low altitudes and vice versa; however, we understand the reviewer's point and have flipped the y-axis back.

7. Line 159: "biases are positive" could be made more direct by replacing it with "the model is warm-biased."
Thank you for this suggestion. We have updated the wording here.

8. Line 166: The phrases "below 100 hPa" and "pressures higher than 100 hPa" are potentially ambiguous because pressure decreases with altitude, so "below 100 hPa" may be unclear whether it refers to pressure or altitude. I recommend making all similar statements consistent, for example by using "pressures higher than 100 hPa." Please apply this consistently throughout, including around line 184.
Thank you for catching this. We have changed the wording to read "pressures greater than 110 hPa" for lower altitudes and "pressures less than 110 hPa" for higher altitudes.

9. Figure 7: The legend for Figures 7a–b does not match the plotted curves.
In Figures 7a-b, POSIDON results are shown in gray, ATTREX in black, and simulations in color, as indicated by the font color. We cannot fit all 8 labels within a single panel, so they are spread across both. In an attempt to further clarify, we add text to the caption that specifies that the colored traces are the ICON simulations in both panels a and b.

10. Line 274: After "Yi et al.," please indicate the corresponding run names.
Here we add "(*M3O)" after the mention of Yi et al. simulations to indicate which runs we are discussing.

11. Figure 9c: Please add a legend.
Thank you for pointing out this omission. We have added one, along with a more detailed description in the Figure 9 caption.

**Reviewer 2 Comments**

**General Comments**

This manuscript presents a comprehensive evaluation of the ICON model at storm-resolving resolution (2.5 km) over the Asian Monsoon region. By using unique high-altitude in-situ measurements from the StratoClim campaign, along with additional datasets (ATTREX, POSIDON, Strateole-2), the authors provide a valuable assessment of the model's performance in the UTLS region. The study successfully links thermodynamic biases (cold/moist UT, warm/dry LS) to microphysical outcomes (ice cloud placement) and dynamical deficiencies (weak vertical velocities and gravity wave activity). The identification

of mechanisms, specifically the underrepresentation of convective overshooting and the potential role of gravity waves, adds significant scientific value.

The manuscript is well-written and logically structured. However, I have noted a few minor typos and instances where figure captions are somewhat unclear or incomplete regarding legend definitions. I recommend publication after minor revisions to address the specific points listed below.

We appreciate the reviewer's time and effort in reviewing our manuscript and for their feedback on our work. Responses are in blue below.

**Specific Comments**

1. Page 1, line 10: The text mentions "frequencies greater than $10^3$ s$^{-1}$". This corresponds to 1000 Hz (acoustic range), which is physically impossible for atmospheric gravity waves. Section 3.3 correctly identifies the range as $f > 10^{-3}$ s$^{-1}$. Please correct the exponent in the abstract to $10^{-3}$ s$^{-1}$.

   Thank you for catching this error in the Abstract. We have corrected the exponent.

2. Page 3, line 68: The text mentions "ICON model, version 2.6.4". Please provide the citation or reference for this model version.

   We have added a reference to Giorgetta et al. *J. Adv. Model Earth Sys.* (2018) here (https://agupubs.onlinelibrary.wiley.com/doi/full/10.1029/2017MS001242). This article focuses on the default model physics for the atmospheric component of ICON, which should be more relevant to understanding our results than other references focused on the ICON dynamical core or grid refinement.

3. Page 3, line 86: There is a discrepancy on the latitude range of the subdomain. The text (Page 3, Line 86) states the subdomain extends from "19 to 30°N", whereas the caption of Figure 1 defines it as "20°N–30°N". Please check and ensure consistency between the text and the figure.

   Thank you for your thorough reading here. The domain listed in the text was correct; we have updated the caption to also read 19-30°N.

4. Figure 1 caption: The text describes the trajectory initiation area as a "gold" box, while the Figure 1 caption refers to it as a "yellow box". Please use consistent wording to avoid confusion.

   Thank you for catching this. We have corrected it.

5. Page 5, line 110: The text refers to the simulation setup as "2M0O", but Table 1 lists the designation as "2M0Ot". Please correct the text to match the designation in the table.

   Here, we have adjusted the wording to read "*We show results only of a 2M0O setup, denoted 2M0Ot, for these online trajectories.*"

6. Figure 3: The caption needs revision to match the panels. Specifically in Panel c:

- (a). It mentions "thin red lines" (which appear orange) but fails to define the solid thick red line (most likely the MLS average).
  We update the color to be "red-orange" and clarify that the bolded trace is the MLS "swath closest to the Flight 7 track" in the Figure 3 caption.

- (b). The color for the ERA-5 reanalysis (purple) is not specified, whereas other datasets are explicitly described. For consistency, this should be added.
  We now specify that the ERA-5 reanalysis is shown in purple for this figure.

  Please change the caption for Figure 5 as well.
  We make the same two changes in the Figure 5 caption as in the Figure 3 one.

7. Page 9, line 170: Please change "pressure" to "pressures".
   Thank you; we have corrected here.

8. Page 14, line 253: There is a missing period (".") at the end of the sentence.
   We have added a period to this sentence.

9. Page 14, line 269: The text says the aircraft samples "no ice clouds" at pressures greater than 160 hPa. However, Figure 9a shows a few black stars (in-situ data) in this area, which means some ice was present. Please rephrase this to be more precise. For example, use "samples negligible ice clouds" or "samples almost no ice clouds."
   Thank you for pointing this out. We update the wording to read "*almost* no ice clouds" for pressures greater than 160 hPa.

10. Figure 9: Please add a legend for panel (c) so the reader can understand what the lines represent.
    Thank you for pointing out this omission. We have added one, along with a more detailed description in the Figure 9 caption.

11. Page 18, line 339 & 340 & 346: The initials used for the authors are not consistent. In "Author Contributions," they are written as "ESA" and "MK," but in "Acknowledgements," they are written as "EISA" and "MKK." Please use the same format in both sections.
    Thank you for catching this. We have changed ESA to EISA. We also note that MK denotes co-author Krämer, while MKK denotes co-author Kupinski. We have ensured that this distinction is corrected across Author Contributions and Acknowledgements.

---

## Author Response (AR2)

Please ensure that the colour schemes used in your maps and charts allow readers with colour vision deficiencies to correctly interpret your findings. Please check your figures using the Coblis – Color Blindness Simulator (https://www.color-blindness.com/coblis-color-blindness-simulator/) and revise the colour schemes accordingly. => Figures 3, 5, 7, and 9.

I have updated colors on the MLS, ERA, and Sonde traces in Figures 3 and 5 so that they are more distinguishable for the Green-Blind/Deuteranopia dichromatic view. The simulation colors (*M*O labels) were chosen specifically to be colorblind-friendly from the outset. When I input Figures 7 and 9 into the colorblind simulator, all panels seem distinguishable to me across the different views, so I have not changed anything here. If the editor has specific concerns on this front, I am happy to update once again.